# Barriers to Educational Inclusion in Initial Teacher Training

Pilar Arnaiz-Sánchez [1], Remedios De Haro-Rodríguez [1], Carmen María Caballero [2,*]
and Rogelio Martínez-Abellán [1]

1    Department of Didactics and School Organization, Faculty of Education, Espinardo Campus,
     University of Murcia, 30100 Murcia, Spain
2    ISEN University Centre, University of Murcia, 30204 Cartagena, Spain
*    Correspondence: carmenmaria.caballero2@um.es; Tel.: +34-868883994

**Abstract:** Initial teacher training within the framework of an inclusive school constitutes a fundamental challenge in order to meet the needs of 21st century society. The 2030 agenda establishes in the target of goal four the need for well-qualified teachers, capable of developing inclusive educational responses to the diversity of the needs of the students enrolled in their centres. The objective of this article is to analyse the perception of the school community regarding the initial training of future teachers in providing an inclusive and quality educational response for all students. The design was qualitative, non-experimental and descriptive. The participants were 78 people involved in teacher training and the educational exclusion–inclusion processes in the region of Murcia (Spain). The information collection techniques were 39 semi-structured interviews and 10 focus groups. The analysis of the information was carried out through an inductive categorization process, classifying the information into different analysis codes with the Atlas.Ti program (V. 8). The results of the work indicate that with regard to inclusive education, there is a clear shortfall in the initial training of teachers, a limitation in the acquisition of competences regarding attention to diversity, the presence of theoretical learning which has limited relevance to practical intervention, and a training approach anchored in models from the past which refers to student deficiencies. This work has the value of providing an insight into the barriers present in initial training from the perspective of all the educational actors involved in the educational system, which has not been sufficiently investigated in this field of study.

**Keywords:** teachers; initial training; inclusive education; competences; training focus

## 1. Introduction

Teacher training is a fundamental element in responding to the new educational challenges posed by the present 21st-century society, since it is one of the key factors in improving the professional competence of every teacher, as it contributes to the development of equitable and quality education [1]. In society, and in increasingly plural and heterogeneous classrooms, ensuring a quality response from the education system involves offering inclusive responses for the success of all students in order to attend to their diverse characteristics and respect differences of gender, social origin, ethnicity and culture [2]. Teacher training is one of the main elements by which the inclusion of all students in the classroom can be established and made a reality and can thus encourage the development of truly inclusive educational centres [3].

Quality education therefore requires the continuous updating of teacher training. Undoubtedly, the importance of the teacher's role and their training has never been as obvious as today, because "teachers are the most significant resource for the effectiveness of educational inputs. The quality of their training is intrinsically linked to the fact that they are the main agents of achieving quality education" [4] (p. 474). For this reason, the education administration must promote and stimulate teacher training processes so that teachers fully participate in them [5,6].

The 2030 agenda, in one of the targets of goal four (which is to guarantee inclusive and equitable quality education and promote lifelong learning opportunities for all), establishes the need for a well-qualified teacher, capable of providing inclusive educational responses to the needs of students enrolled in educational centres [7]. Primary and secondary education professionals trained in the inclusive education paradigm are needed, both in initial and ongoing training, who are characterized by having an inclusive profile that leads them to consider inclusion as an essential value in educational centres [8]. Inclusive education is constituted in a systematic process that channels specific values into action and represents the desire to overcome the barriers that lead to the exclusion of many students in order to promote participation and learning [9,10].

We can define barriers as all those contextual factors that hinder or limit full access to education and learning opportunities for children and young people. They appear in relation to their interaction within different contexts: social, political, economic, institutional and cultural. Concerning the identification of different barriers in the development of inclusive processes, Booth and Ainscow [11], indicate those that hinder access to learning and participation, or, as Echeita [12] indicates, others that appear in the school context itself, such as: school culture, organization, the education programme of the centre, curricular aspects, classroom methodology, the family context and the local context.

Victoriano [13], indicates that it is the facilitators and barriers that act as variables that influence inclusive processes. Therefore, the facilitators would be related to those actions that allow or make possible the successful implementation of inclusion; while barriers can be understood as possible difficulties experienced by teachers and students, due to external factors, such as the organization and operation of educational establishments [14]. Barriers are also those things that prevent or hinder the participation of students and teachers, while the factors that make it possible to optimally achieve inclusion are called facilitators [15].

Consequently, there is a need for teachers to understand the meaning of inclusive education, to master inclusive practices and understand what inclusive pedagogy entails [16] so that all of them become facilitators of inclusion and not barriers to it [17]. In this way, teachers will be more capable of creating inclusive learning environments, which will contribute to breaking down the barriers that may have existed in their training [18,19]. In this regard Orozco and Moriña [20] establish an ideal teacher profile based on personal and professional skills such as: "responsibility, reflection on practice, supportive relationship, respect and empathy for students, use of didactic approaches, teaching with real practical examples, careful planning and passion and enthusiasm for the profession" (pp. 2–3).

For all these reasons, initial teacher training should consist of a general component and a professional component. The first component refers to the theoretical and scientific knowledge of the set of subjects that the teacher is going to teach in the future, while the second component refers to the acquisition of skills specific to the teaching profession and practical training in schools [21]. Through the second component, future teachers will acquire the psycho-pedagogical and didactic skills necessary to practice their profession [22]. Depending on how both components are combined, we can find two initial training models [23,24]: a simultaneous model that combines general and professional training in the same curriculum. The training of the subjects that the teacher is going to teach in the future is combined with pedagogical and didactic studies, including internships in schools, which promotes the relationship between theory and practice. Additionally, a consecutive or successive model where future teachers first receive training in the disciplines they will teach in the future (general training) and, later, pedagogical and didactic training (professional training), which will allow them to practice as teachers.

According to UNESCO [25], initial teacher training must be based on principles of inclusion and equity. Teachers must learn teaching methods that include all students and must know the mechanisms of exclusion and discrimination. For this reason, there is an aspiration to create an inclusive curriculum that meets the expectations and needs of all students in heterogeneous learning environments. This curriculum must be correlated with transversal issues such as citizenship, human rights, gender equality, a culture of peace and

non-violence and sustainable development; these themes must play a leading role in study plans and programmes [26,27]. Forlin [28] indicates that inclusive education directly affects the value system of teachers, questioning their most intimate beliefs about what is correct and fair. Preparing new teachers to be "inclusive" requires much more than simply adding a special education course or module, and teacher trainers must develop the expertise to address conflicting issues and deal with their own deeper values and attitudes.

The European Agency for the Development of Education for Pupils with Special Educational Needs [29,30] determined some of the competencies that teachers should possess to promote and work for inclusive education. These are: valuing and supporting the progress of all students, teamwork, using various teaching methods, promoting active and participatory learning experiences, and diversifying teaching content and assessment methods. However, currently, there are two positions regarding the training that teachers should receive in order to teach in inclusive environments [31]. On the one hand, there are those who defend the position that there should be a greater focus on the knowledge of the different types of difficulties that some students may present and the teaching strategies to work with them. On the other hand, there is the position that defends that inclusion is an opportunity to rethink the functioning of the school and for professionals in the educational field to make a critical reflection on their teaching methodology and their beliefs about these differences and thereby reflect on their own way of working [32].

The research carried out in Spain by Sánchez-Serrano et al. [33] on training for inclusive education in primary education teacher degrees, where the training programme of 39 public universities was studied concludes that the training received by students who are studying for these degrees is insufficient. The non-obligatory nature by which Order ECI/3857/2007 allows the incorporation or non-incorporation of specific subjects of inclusive education in the study programme results in a meagre presence of the same in them and, therefore, deficiencies in the training of future teachers of primary education. Hence, it would be beneficial to adopt changes in this regard, which has already been recommended to Spain by the European Union, as well as the universities adopting the commitment to apply the inclusion paradigm as a transversal axis of initial teacher training.

In relation to the above, in the study by Muntaner-Guasp et al. [34] in which the study plans of Spanish universities were analysed, it was shown that the training offered to teachers at the university stage focuses on an integrating model—or deficit model—focused on the categorization of the students and looking for specific answers for students with specific difficulties. Therefore, it is a formative approach anchored in the past, inconsistent with an education for all model, which implies insufficient training for teachers and a departure from the paradigm of inclusive education. According to this study, this situation makes an equitable quality education for all impossible and must be alleviated. Thus, education programmes should focus on models or approaches based on intervention, on making teaching and learning processes more flexible, and not on labelling students based on their characteristics. In addition, the evaluation of the attitudes of the teaching staff must be real and deep, in order to achieve a modification of study plan programmes and carry out real inclusive practices in educational centres.

In the international context, the study by Ritter et al. [35], analysed the beliefs of pre-service teachers with regard to inclusive education. Their results showed that pre-service teachers who worked in multi-professional teams improved their knowledge of inclusive education through the incorporation of facets such as individualization or differentiation, while pre-service teachers who worked in mono-professional teams did not demonstrate such learning. Another notable international study is that of Ritter et al. [36]. This paper analysed the effect of mono-discipline compared to multi-discipline collaboration on the attitudes of future teachers towards inclusive education. In relation to attitudes towards inclusive education, the study showed that participants belonging to "multi-disciplined teams" acquired a more positive change, while these results are not acquired in mono-disciplined participants.

Therefore, the purpose of this study focuses on analysing the perception of the school community of the initial training of teachers in providing an inclusive and quality education response to all students in early childhood, primary and secondary education centres. For that purpose, the research questions of this study are:

- Are there barriers in initial teacher training when it comes to acquiring skills for inclusive education?
- Have teachers received professional or practical preparation in their training to meet the diversity of student needs?
- What training approach prevails in teacher training regarding attention to diversity?

## 2. Methods

### 2.1. Design

This research has been developed with a qualitative, non-experimental and descriptive design [37]. A qualitative design has been selected since it affords the opportunity to establish a framework in which a dialogue model is identified between the various educational actors participating in the study [38]: teachers, counsellors, families, students, management teams, unions, associations and political representatives. In this way, we achieve a better understanding of the experiences and life lessons, the personal situations and the perceptions towards the various aspects that are the object of the study being carried out.

### 2.2. Participants

The participants of this research were 78 people involved in the processes of educational exclusion–inclusion and with teacher training in the Region of Murcia (Spain). Specifically, these people belong to the following groups: 5 members of a management team of educational centres; 3 representatives of the teacher training centre (TTC); 36 teachers (12 from nursery and primary schools, 11 from secondary schools and 13 specialist teachers in attention to diversity); 14 students (4 university students from the primary teacher's course, 4 university students from the infant teacher's course, 2 master's students in teacher training and 4 secondary school students with special education needs); 7 family members; 2 association representatives; 4 union representatives and lastly, 7 education counsellors (see Figure 1):

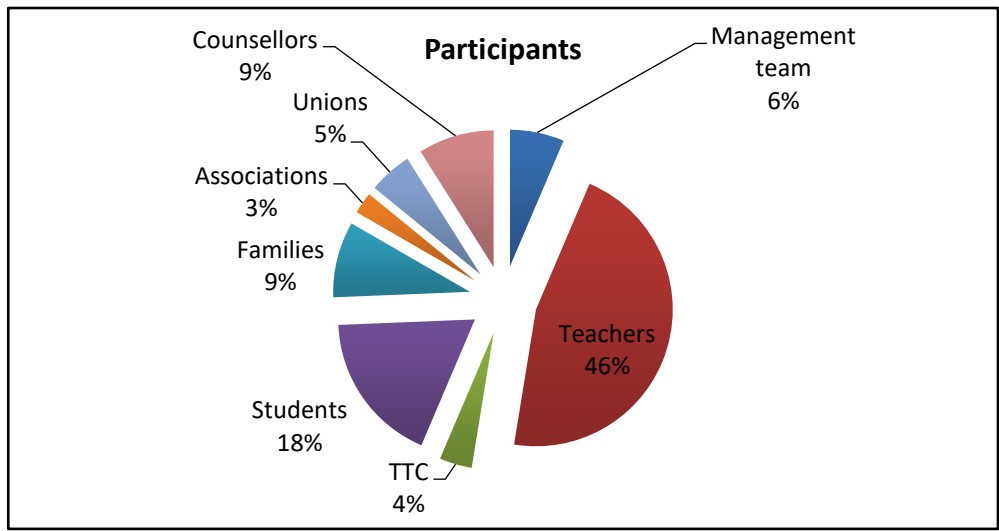

**Figure 1.** Participants in the study by groups. Source: Prepared by the authors.

The participants in the focus groups and in the interviews were personally invited to participate in the study, so it was an intentional selection of the sample until information saturation was achieved. To do this, a personal invitation letter was sent to each of

them using email, or they were contacted by telephone. An attempt was made to have the participation of people who, although they belonged to the same group, presented heterogeneous characteristics in relation to educational stage, gender and geographical origin. In this way, it was possible to obtain information according to the different realities experienced in the educational centres in the Region of Murcia.

*2.3. Information Gathering Techniques*

For the collection of information, two qualitative techniques were used:

- Semi-structured interviews: A total of 39 semi-structured interviews were carried out. These were carried out by researchers from the group "Inclusive Education: A School for All" (EDUIN) with: 5 centre directors; 7 educational counsellors; 1 teacher; 3 family members; 10 students; 6 political representatives, 5 trade union representatives and 2 association representatives. Some of the interview questions were: Are there barriers in initial teacher training to inclusive education? Do the university training centres offer teachers the necessary skills and abilities to attend to diversity? Are the teachers aware of the educational legislation for attention to diversity? What formative approach is used in universities in relation to inclusive education? What role should the educational administration assume in the promotion of inclusive education in teacher training?
- Focus groups: A total of 10 focus groups were conducted, 7 of them with teachers, 1 with families, 1 with students with disabilities and 1 with professionals from the Teacher Training Centre (TTC). To carry out the focus groups, a guide was designed in which information was collected regarding: 1) instructions and contextualization of the study; 2) socio-demographic data of the participants; and 3) topics of interest on teacher training in the Region of Murcia regarding attention to diversity (initial training and ongoing training).

The script for the questions, for both the focus groups and the interviews, was validated by an inter-judge process. Likewise, all participants were asked to provide informed consent in which they confirmed their explicit wish to voluntarily take part in this research.

*2.4. Process*

To carry out this research, we have the approval of the Research Ethics Commission of the University of Murcia. In this study, the following development phases have been followed:

1. Documentary analysis phase: In this phase, we analysed reports, regulations, news and scientific articles referring to the provision of an inclusive education for all students in the initial and permanent training of teachers. This phase allowed us to recognize the reality of the current situation and elaborate the theoretical framework of the present investigation. Likewise, it provided clear evidence of the research problem under study and helped to specify the specific objectives of this study.

2. Collection and analysis of qualitative information phase: In this phase, contact was made with the participants: an email with an invitation letter was sent to the possible participants in the study through the management of the educational centres or via phone. In the email, information regarding the project was included, what the procedure would be and the objective of the investigation. Then, we proceeded to conduct the interviews and focus groups. Once the information was obtained, the content was transcribed and the transcription documents were added to the "Hermeneutical Unit" of the qualitative analysis program. Finally, the inductive analysis of all the data was performed.

3. Information triangulation phase: In this phase, the information obtained in the interviews and focus groups with the participants was triangulated together with the most relevant data from reports, regulations, news and scientific articles from the previous documentary analysis. All this has facilitated the elaboration of the discussion of the study, offering a realistic overview of the objective of this study: barriers to the progress of inclusive education in the training of teachers.

### *2.5. Analysis of the Information*

For the analysis of the data, a verbatim transcription of the information collected in the interviews and in focus groups was made, followed by an analysis of its content. For this, the qualitative statistical support software ATLAS.ti (The Qualitative Data Analysis and Research Software, version 8 for Windows) was used. The content analysis has been carried out following an inductive model, in which the categories of analysis and the codes for the classification of the information (textual citations) were established [39,40]. Presented below is the system of categories and codes for classification (See Figure 2):

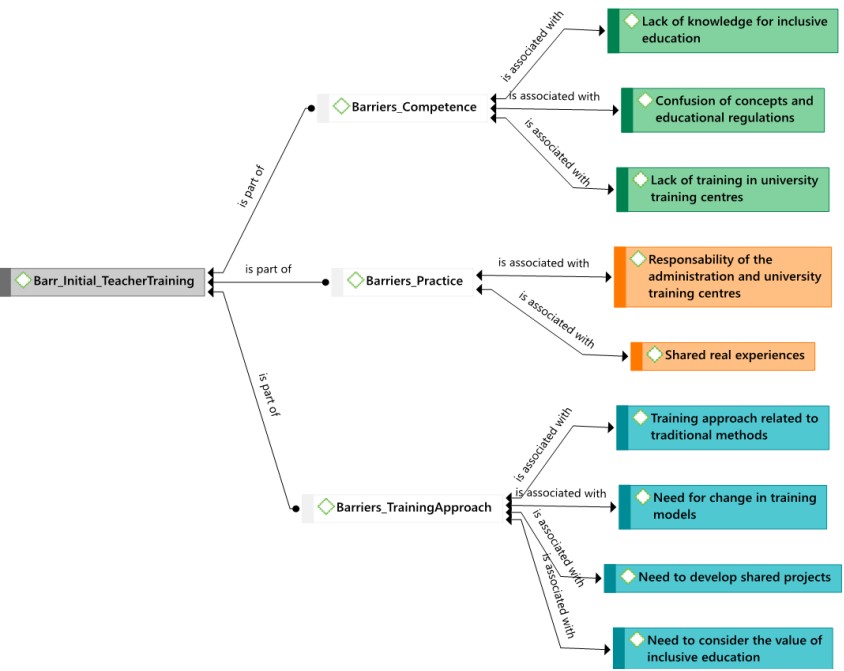

**Figure 2.** Codes for classification. Source: Prepared by the authors.

## 3. Results and Discussion

The total number of citations that allude to barriers in initial teacher training has been N = 79 citations. Its distribution by groups of participants is detailed in Figure 3.

| | | Barr_Initial_TeacherTraining 80 |
|---|---|---|
| ● ◇ ASSOCIATIONS | 130 | 3 |
| ● ◇ COUNSELLORS | 310 | 16 |
| ● ◇ TTC | 36 | 1 |
| ● ◇ FAMILIES | 196 | 3 |
| ● ◇ MANAGEMENT TEAM | 224 | 4 |
| ● ◇ STUDENTS | 174 | 9 |
| ● ◇ TEACHERS | 531 | 35 |
| ● ◇ UNIONS | 102 | 8 |

**Figure 3.** List of citations by group of participants: "Initial Teacher Training Barriers". Source: Prepared by the authors.

The interviews and focus groups carried out have revealed the existence of important barriers in the development of inclusive education in the initial training of future teachers. Below we present the textual information given in response to the following formulated research questions.

*3.1. Are There Barriers in Initial Teacher Training Regarding the Acquisition of Skills for Inclusive Education?*

The participants point out a lack of acquisition of the necessary skills to respond to the diversity of the student body within the framework of an inclusive education. These training barriers have been highlighted by different groups, indicating the poor preparation of teachers to respond to the diversity of students and towards sounder skills [13,14]. This coincides with the studies carried out by Sánchez-Serrano et al. [33] and by Muntaner-Guasp et al. [34], where insufficient training in inclusive education within the curricula of Spanish universities is revealed.

> I would add that there are very few hours of education dedicated to inclusion in university studies. I do not know about you, but I did not do any (Secondary_Teacher).

The result of this is that in many cases, new teachers, when they arrive at the centres, manifest a lack of skills that leads them to confuse concepts, since they do not know the theory of inclusive education very well, nor the current legal regulations in the educational centres. Likewise, it shows ignorance of the specific measures for attention to diversity. This teacher profile is far from the ideal profile that Orozco and Moriña [20] establish when they set forth the personal and professional skills that teachers should have. In turn, it denotes a barrier in the acquisition of the competences that they should have as teachers [7], and that this still occurs today is worthy of attention. To avoid this situation, Ritter et al. [36], believe that a pre-requisite for competent teachers is that they should be well prepared during their initial training on the meaning and conceptualization of inclusive education. These authors have verified that if this training process is carried out, it will bring about a change of attitudes in future teachers, which they consider essential for putting inclusive education into practice and achieving success for all students.

> Graduates fresh out of university do not know how to distinguish between a Student with Specific Educational Support Needs (ACNEAE) and a Student with Special Educational Needs (ACNEE). They do not know what he decrees re diversity is, they have no idea what measures should be applied... They are absolutely stuck on something as fundamental as that. Additionally, I find myself repeating the same thing again and again, forty years ago when I was at university, it was considered normal, but today? (Primary_Teacher).

This opinion is confirmed by other researchers [17], when they state that many teachers do not feel well enough prepared to meet the needs of all students in a heterogeneous class, conscious of the fact that they sometimes act by trial and error. Thus, the participants in this study consider centres of higher education as one of the barriers to inclusion, given the scant training they consider is provided to students and future professionals. This situation has been pointed out by Friesen and Cunning [4], when they state that teacher training is essential in the instigation of the development of truly inclusive centres and classrooms.

> There is a need for better training at the University level. They leave university unclear about many things, even in the area of specialization, special education. That is essential (Families_Secondary).

As can be seen, this barrier becomes more pronounced if we refer to the secondary education stage. In these studies, knowledge of the subjects studied is prioritized more than its pedagogical application. This, undoubtedly, creates a difficulty in achieving inclusive, equitable, participatory and quality teaching processes in the classrooms [1,2,4].

Additionally, if it is already a problem in infant and primary education, which is already an obstacle, it also happens with secondary school and sixth form teachers [...] These people have studied a degree in mathematics, physics, chemistry, where there has been no subject related to the world of education, neither didactics nor organization; absolutely nothing. [ . . . ] Their academic curriculum should be changed, and whoever wants to be a teacher should take some electives that will direct them towards the world of education (Management team_Primary).

In my opinion, there is a lack of training on the part of future teachers regarding inclusive education, starting with their principal degree subject and continuing through to the Master's Degree in Teacher Training (in which inclusion is given only superficial attention) (University Students).

*3.2. Have Teachers Received Professional or Practical Training in their Training to Meet the Diversity of Student Needs?*

The participants comment on the virtual absence of training linked to inclusive educational practice during initial teacher training, which would prepare teachers to respond to heterogeneous characteristics of the students present in the classrooms, as already verified in other works [16,18,19]. In the opinion of Ritter et al. [35], a good method of linking theory with practice is to offer future teachers practical experiences in inclusive classrooms. This would give them the opportunity to confront their beliefs with reality and reflect on how their initial training is progressing. The results obtained by these authors indicate that this formative process improves the beliefs and abilities of these students.

There is a very large gap between the university and the real life of educational centres. One thing is theory and another thing is day-to -day practice (Primary_Teachers).

I left teacher training without knowing how to teach a child to read, and when I finished my third year I said: "Okay, now I am going to a classroom and I do not know how to teach a child to read" (Counsellors).

In university degrees, only very general training is given on attention to diversity. This does not correspond to the real difficulties that exist in schools and classrooms (Unions).

It is understood that education administration and centres of higher education should be the bodies in charge of preparing teachers to put into practice current educational approaches and models [5,6]. Despite this, in this research, the participants point out that there are limitations in teacher training that make it difficult to develop effective inclusive practices in the methodological approaches required by the inclusion paradigm.

With the changes, in the new methodological criteria that they want to implement, the first problem arises, which is the initial training of teachers, since there should be a much greater connection between what is learnt at university and what is actually applied in the classroom (Management Team_Primary).

This fact is ratified by the participants who are currently finishing their studies, when they point out that the knowledge they have obtained is not useful in confronting the reality of school life, since it has not allowed them to acquire sufficient skills to promote an inclusive education. These opinions contrast with the training indicated by the European Agency for the Development of the Education of Pupils with Special Educational Needs [29,30], regarding the profile that teachers must acquire to promote inclusive education in schools.

In my case, the theory given at the university has not served me as much as I imagined. Let me explain: despite the fact that the contents acquired are real and useful, it is not until you face reality in an education centre (in 4th grade) that you really understand everything that was only previously explained on paper (University Students).

*3.3. What Training Approach Prevails in Teacher Training Regarding Attention to Diversity?*

The participants indicate the existence of a focus on initial teacher training linked to old models where specialities reigned, or "areas of specialisation" as they are currently called. Thus, the actors participating in this study point out that teachers who have specialized in Therapeutic Pedagogy or Hearing and Language Support, through these areas of specialization, are better prepared to provide an inclusive response to students with difficulties. Explicitly, teachers should move away from this training position [31,32], and look for training models in which all professionals commit to promoting appropriate teaching methods for all students, as well as inclusive curricula that meet the heterogeneous learning needs of all students [26,27].

> We would have to start by offering different training to all teachers (Management Teams_Primary).

> I sincerely believe that most teachers were not well prepared to care for me. The only ones who have known how to teach me well have been the teachers of Therapeutic Pedagogy and Hearing and Language Support in the open classrooms (Student_Secondary).

> For example, many teachers do not know how to teach a child with autism because they have not studied subjects that teach them how to respond to the needs of this type of student. There should be subjects or courses on attention to diversity for all teachers in university training -and not only for teachers who study the specialist degree in Therapeutic Pedagogy—(Associations).

Consequently, higher education centers are obliged to change their training models in order to respond to this reality and not remain anchored in obsolete models that do not respond to the needs present in the today's classrooms [21,22]. This reality has been denounced by the participants in this study, highlighting the need to modify this approach.

> That curriculum should be changed and whoever wants to be a teacher should take some electives that would direct them towards the world of education. Additionally, you no longer need a master's degree, since within your academic curriculum you study an education module. However, well, that a physicist who is extremely intelligent gets into a classroom of the first or second stage of Compulsory Secondary Education (ESO)... Let us see what he does with thirty-five students of that kind (Management Team_Primary).

> The theory fails us when other teachers later tell us, very vehemently: "It's because that boy in the class is not going to learn, there's no way, I cannot do anything for him..." Well, you believe it, you believe it because you lack the spark to say: no, there has to be another way of doing it (Teacher_Primary).

In this regard, it is worth noting the multi-professional co-teaching model that is being developed in some countries such as Germany [35]. This study has shown that if future teachers experience co-teaching in their initial training, through multi-professional teams, they improve their knowledge of attention to diversity and, therefore, expand their skills on inclusive education. These authors conclude that multi-professional co-teaching helps future teachers to acquire a successful training based on inclusive education.

Inclusive education also needs to be considered as an essential value in educational centres and in the planning of teacher training. In this sense, Arnaiz-Sánchez [9], Forlin [28], Gavish [8], and Waitoller and Thorius [10] express the importance of taking into account the implicit values in inclusive education, considering this as a key factor for the presence, participation and learning of all students in schools.

> We are clear that the initial teacher training would have to undertake an important change of direction. However, well I think it has to come from above. [ . . . ] We are in a country where I believe that education does not matter to anyone. What matters in education? The commercialization of education. Additionally, I say

it again, it does not matter to anyone, because if it mattered things would have changed (Teacher_Secondary).

Hence, the need to develop participatory projects shared among the entire school community and to encourage cooperation between different professionals, developing truly inclusive educational scenarios [4].

It is very important to promote innovative projects and the development of collaborative learning communities between teachers (Teacher Training Centre).

The aforementioned demonstrates the lack of training that future teachers receive in their passage through centres of higher education in order to respond to the diversity of the student body and offer a quality, equitable, and therefore inclusive education to all students. In this way, the participants demand more practical training linked to the socio-educational needs of the centres; not only this, but it is also directly related to the proposals of Sustainable Development Goal Four present in the 2030 Agenda [7]. In order to achieve this, training must be managed by the whole teaching team, preparing and making everyone responsible for its promotion.

### 4. Conclusions and Implications for Education

Achieving inclusive education inevitably entails training teachers and equipping them with the necessary skills to face the challenges present in the 2030 Agenda [1]. Therefore, in order to promote equitable and quality education, we require teachers trained in the inclusive education paradigm which will provide them with the skills to adopt it as a belief and as an intervention practice in educational centres.

This study reveals the existence of significant deficiencies in the initial training of teachers, demonstrating a discrepancy between education administration and centres of higher education, bodies responsible for preparing teachers to put into practice current educational models and approaches. Specifically, the results of this study show the need to stimulate and develop these training processes in order to have teachers prepared and willing to face the challenges of inclusion.

The lack of adequate initial teacher training is one of the main barriers to inclusion, and this is reflected in this research. If teachers are not trained to meet the heterogeneous needs of students, they will not be able to develop, day-by-day, inclusive education in the classrooms of education centres.

The different groups that have participated in this research are in agreement in highlighting the lack of initial teacher training needed to develop an equitable and quality education for all.

The current initial training of future teachers does not allow us to get to know the true significance of the inclusion paradigm, which resides in identifying the present barriers to inclusion in order to promote participation and learning for all. This situation entails significant training deficiencies from the start of a teaching career, which can lead to poor practice and the reproduction of old approaches linked to the deficit model, thus perpetuating segregation and exclusion.

In short, and in an attempt to summarize the answers to the three formulated research questions, this paper concludes by stating:

- The existence of training barriers in the initial training of teachers in centres of higher education means that students do not acquire the necessary skills to develop an equitable and quality education for all. In this way, graduates do not have the skills, abilities, knowledge and attitudes necessary to practice the teaching profession from an inclusive approach. This constitutes an important barrier to inclusion.

For these reasons, it would be advantageous to expand the research into the planning of teacher training currently existing in Universities regarding the teachers acquisition of skills and competencies needed for an education for all.

- The presence of barriers is linked to training models based on theory and without a connection to education practice. This undoubtedly, hinders future teachers by failing

to make the connection between theory and practical intervention, and thereby lacking an inclusive pedagogy in their repertoire of skills. The educational administration occupies an essential role in this regard, which it is carrying out inadequately in facing up to the current challenges.

Likewise, the role of the educational administration in the development of training programs which allow for theoretical–practical training, applied to the context of the education centers and to the reality of the classrooms, should be analyzed in greater depth.

- There is a need to rethink the existing teacher training model. When faced with the current challenges, all teachers, both general and specialist, should receive training that enables them to meet the objectives set out in the inclusion paradigm. This requires the abandonment of training models anchored in the deficit model that has characterized education practices for attention to diversity for a long time. Consequently, future research should consider inclusion as an essential value in training plans, as this is the essential principle for an efficient and quality school for all.

For all these reasons, the initial training of future teachers must be rethought by centres of higher education in order to materialize Sustainable Development Goal Four of the 2030 Agenda, which is "Guarantee an inclusive, equitable and quality education and promote opportunities lifelong learning for all".

As a limitation of this current study, we point out that it has been carried out in only one region of Spain, which leads us to treat the results obtained with caution and not to generalize them within the context of national or international realities. However, with a different methodological design from that used in the present investigation, similar results have been obtained in other studies.

**Author Contributions:** Conceptualization, P.A.-S., R.M.-A. and C.M.C.; methodology, C.M.C.; formal analysis, R.D.H.-R. and C.M.C.; investigation, P.A.-S., R.D.H.-R. and C.M.C.; resources, P.A.-S. and R.M.-A.; data curation, C.M.C.; writing—original draft preparation, R.D.H.-R.; writing—review and editing, P.A.-S. and C.M.C.; visualization, R.D.H.-R.; supervision, P.A.-S.; project administration, P.A.-S. and R.D.H.-R.; funding acquisition, P.A.-S. and R.D.H.-R. All authors have read and agreed to the published version of the manuscript.

**Funding:** Research project: "What are we forgetting in inclusive education? A participatory research in the Region of Murcia". Funder: Ministry of Science and Innovation of Spain, Funding Number: PID 2019-108775RB-C44-EDU.

**Institutional Review Board Statement:** The study received a favorable report from the Research Ethics Commission of the Universidad de Murcia (Spain).

**Informed Consent Statement:** Informed consent was obtained from all subjects involved in the study. In addition, the confidentiality of the participants was guaranteed by use of numeration (with respect to interviews and questionnaires).

**Data Availability Statement:** The data presented in this study are available on request from the corresponding author. The data are not publicly available for privacy reasons.

**Acknowledgments:** The authors of this study would like to thank the educational community for their participation in this research.

**Conflicts of Interest:** The authors declare no conflict of interest.

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
