# Peer review of "Barriers to Educational Inclusion in Initial Teacher Training"

_societies, doi:10.3390/soc13020031_

Round 1

Reviewer 1 Report

On a general level, the topic addressed in this paper is of interest to the field of the journal and to the research community. Indeed, teacher education in the framework of an inclusive school is a current and very important topic in the society of the 21st century. However, there are some small issues that need to be modified in this study. These are presented below according to the different sections of the study.

Abstract:

It is suggested to answer these questions at the end of the abstract to highlight the value of the article presented: what is special about this study from those that already exist in this field? In other words, what is its originality or value?

1. Introduction    

Lines 116-138 discuss two studies that have been carried out at national level with reference to the field of study. But what about international developments in inclusive education in other countries? Similarly, some international studies should also be mentioned.

Regarding the research questions posed, the last one "And what training approach prevails in teacher training regarding attention to diversity?" should be corrected, as it is not possible to start a sentence with "And" and therefore this word should be omitted.

Finally, there is a typographical error in the section, as there is an extra space in line 63 between "or" and "hinder".

2. Methods

2.1. Design

It is stated that a qualitative design has been used as it offers the opportunity to establish a framework in which a model of dialogue between the different educational actors involved in the study is identified and, furthermore, that this design achieves a better understanding of life experiences and learning, personal situations and perceptions towards the various aspects that are the subject of the study conducted. Which author(s) state this? Although reference is made to the research design, its choice must be justified in terms of the authors. 

2.2. Participants

This section specifies the group to which the participants belong. The following questions arise in this regard:

- On a specific level, what do the acronyms CPR refer to in line 163? It is an acronym that refers to a terminology used in Spanish "Centro de Profesores y Recursos de la Región de Murcia" and therefore should be specified, as it does not fit with the term used in English "teacher training centre".

- It is also stated that "36 teachers" have participated. What stage of education are the teachers from? It should be specified whether they are pre-school, primary, secondary, or special education teachers, etc. In this way, it will be possible to specify whether it is a representative sample, as discussed below.

- Similarly, "14 students" is mentioned. What are they students of? It should be specified whether they are students of the Degree in Early Childhood Education or Primary Education; or if they are students of the Master's Degree in Education, or Educational Counselling...; or if they are secondary school students...

The figure presented is framed on all sides except for the top part. The format should be revised. On the other hand, there are errors in the English language, since when reference is made to "education counsellors" the term "councillors" is used, which is incorrect, since it refers to another public figure. In addition, the term "estudents" is used, which is misspelled, as the correct spelling is "students". Similarly, the acronym "CPR" should be revised as mentioned above.

2.3. Information gathering techniques

As for the semi-structured interviews conducted, examples of questions that were asked should be given for the purpose of gaining more in-depth knowledge of the study.

2.5. Analysis of the information

Regarding figure 2, it contains information in Spanish such as "es parte de" and "está asociado con". The language should be revised and corrected.

3. Results

Overall, the presentation of the results is correct, as they are presented on the basis of the research questions posed. However, they are linked to the discussions of the study. In order to achieve a better quality of the study presented, it is recommended to separate the discussions from the results and to create a new section called "Discussion", where the results are interpreted, looking at the causes and comparing them with other studies. Specifically, in the results section, it should simply be the presentation and description of the results.

Similarly, it is recommended that when detailing a response from "university students", the degree or Master's degree that these students are studying should be indicated, so that the opinion they are conveying can be understood in greater depth.

It is also recommended that some of the answers given by the "associations" and "unions" be added, as even though they have taken part in the study, no opinion from these groups can be found with respect to any of the research questions.

Figure 3 should be placed in the centre of the text, as are the rest of the figures. Likewise, the acronym "CPR" should be revised, as mentioned above, and replaced, for example, by the words "teacher training centre".

Finally, the numbering of the subtitles should be revised. The results section corresponds to number 3, but the subtitles are numbered as number 4, and this should be corrected.

4. Conclusions and implications for education

Regarding the conclusions and practical implications, it is recommended to add at the end of the paragraphs already written, the future lines of research and the practical implications of the present study in the current educational panorama, highlighting the continuation of the study.

References

There are a few typographical errors that need to be corrected:

In reference 9 there is a missing full stop after the pages and before the DOI.

In reference 11 there is an extra space before "Teachers".

In reference 21 the authors should be separated by a semicolon ";".

In reference 29 there is a red full stop.

Author Response

Dear,

Regards.

Reviewer 2 Report

Dear, 

From my perspective, I consider that this article meets the criteria of pertinence, quality and relevance. It has all the components to be categorized as a scientific article. In addition, the topic addressed is of great importance for the field of inclusive education. 

In general, it is a well-structured article and is ready for publication. My only suggestion is to use bibliographical references to support everything related to the selected methodological design. 

Congratulations!

Author Response

Dear, We have incorporated references to justify the design used.
Please see attached document. Regards.

Reviewer 3 Report

The manuscript "Barriers to Educational Inclusion in Initial Teacher Training" deals with a very important topic and prerequisite to realize inclusion in schools. The article is well written und straightforward, however I have some comments on it:

Introduction: as you analyze the situation in Spain, you mainly refer to sources of Spanish scientific studies. However, if this article is to be published internationally, it would be advisable to also include international sources.

Methodology: this section is sound and stringent. One minor thing: the labelling in Figure 1 is not throughout in English (Estudents)

Results: the above is also true for Figure 2. Otherwise, This section is well done

Conclusion and implications: To be honest, I miss a discussion of the findings. How do your findings relate to similar findings in other areas/countries? What are examples for teacher preparations for inclusion internationally? How do your findings relate to the requirements? Furthermore, it would be advisable to not only call for a change in teacher preparation, but also to suggest interventions that proved successful (e.g. to be found in Ritter et al. (2020). Effect of same discipline compared to different discipline collaboration on teacher trainees’ collaboration skills and attitudes towards inclusive education. Teaching and Teacher Education or Ritter et al. (2019). Pre-service teachers’ beliefs about inclusive education before and after multi- compared to mono-professional co-teaching: An exploratory study. Frontiers in Education- Teacher Education).

Author Response

Dear, 

Regards.
